# Clinically Evaluated COVID-19 Drugs with Therapeutic Potential for Biological Warfare Agents

**DOI:** 10.3390/microorganisms11061577

**Published:** 2023-06-14

**Authors:** Ido-David Dechtman, Ran Ankory, Keren Sokolinsky, Esther Krasner, Libby Weiss, Yoav Gal

**Affiliations:** 1Pulmonology Department, Edith Wolfson Medical Center, 62 Halochamim Street, Holon 5822012, Israel; ido.dechtman@gmail.com; 2Sackler Faculty of Medicine, Tel Aviv University, Tel Aviv 6997801, Israel; 3The Israel Defense Force Medical Corps, Tel Hashomer, Ramat Gan, Military Post 02149, Israel; ranankory@gmail.com; 4Chemical, Biological, Radiological and Nuclear Defense Division, Ministry of Defense, HaKirya, Tel Aviv 61909, Israel; kerensokolinsky@gmail.com (K.S.); esther_krasner@mod.gov.il (E.K.); 5Department of Biochemistry and Molecular Genetics, Israel Institute for Biological Research, Ness Ziona 74100, Israel

**Keywords:** SARS-CoV-2, COVID-19, treatment, drugs, antiviral, host-directed therapy

## Abstract

The severe acute respiratory syndrome coronavirus 2 (SARS-CoV-2) outbreak resulted in hundreds of millions of coronavirus cases, as well as millions of deaths worldwide. Coronavirus Disease 2019 (COVID-19), the disease resulting from exposure to this pathogen, is characterized, among other features, by a pulmonary pathology, which can progress to “cytokine storm”, acute respiratory distress syndrome (ARDS), respiratory failure and death. Vaccines are the unsurpassed strategy for prevention and protection against the SARS-CoV-2 infection. However, there is still an extremely high number of severely ill people from at-risk populations. This may be attributed to waning immune response, variant-induced breakthrough infections, unvaccinated population, etc. It is therefore of high importance to utilize pharmacological-based treatments, despite the progression of the global vaccination campaign. Until the approval of Paxlovid, an efficient and highly selective anti-SARS-CoV-2 drug, and the broad-spectrum antiviral agent Lagevrio, many pharmacological-based countermeasures were, and still are, being evaluated in clinical trials. Some of these are host-directed therapies (HDTs), which modulate the endogenic response against the virus, and therefore may confer efficient protection against a wide array of pathogens. These could potentially include Biological Warfare Agents (BWAs), exposure to which may lead to mass casualties due to disease severity and a possible lack of efficient treatment. In this review, we assessed the recent literature on drugs under advanced clinical evaluation for COVID-19 with broad spectrum activity, including antiviral agents and HDTs, which may be relevant for future coping with BWAs, as well as with other agents, in particular respiratory infections.

## 1. Introduction

The severe acute respiratory syndrome coronavirus 2 (SARS-CoV-2) is the causative agent that led to the outbreak of the COVID-19 (Coronavirus Disease 2019) pandemic, which resulted in approximately 660 million infected and approximately 6.7 million deaths worldwide as of the end of December 2022 (https://www.worldometers.info/coronavirus/, accessed on 31 December 2022). This is most likely an underestimate, and according to the WHO (world health organization), the numbers might be much higher. In the first two years of the pandemic (2020–2021), WHO estimates the excess mortality (including non-reported mortality) to be 14.83 million people, 2.74 times the actual reported cases for this period (5.42 million) [1].

The virus affects the respiratory system and, in some cases, leads to severe lung damage characterized by a violent inflammatory response manifested by overexpression of pro-inflammatory cytokines and other indicators of tissue damage (“cytokine storm”). Although the development of the inflammatory response is a critical mechanism of the innate immune system for coping with viruses and other pathogens, the aforementioned excessive activation of those mechanisms might in many cases lead to pathological results, mainly severe tissue injury and even death [2].

Although the vaccines developed against SARS-CoV-2, particularly the mRNA-based vaccines that underwent swift emergency approval process, are the ultimate tool in the prevention of disease development and reduction of its severity, a significant part of the world’s population has yet to be vaccinated or exposed to the virus. In addition, in a small but non-negligible part of the population, the vaccine is not effective (such as immunocompromised patients). Furthermore, the immune response fades over time, necessitating repeated vaccinations, and vaccine-resistant variants, such as the Omicron and its derivatives, have also emerged [3]. All of these reasons may explain why there are still many documented cases of severe morbidity and even mortality associated with the COVID-19 disease, necessitating the provision of additional treatments and medications. 

The COVID-19 pandemic has catalyzed significant advancements in medical research, including the development of diagnostic tools, rapid identification methods of new variants, and widespread administration of mRNA vaccines. Additionally, numerous studies have been conducted to investigate the potential of existing and novel drugs to treat patients, both for individuals with severe illness and those in the community.

This review focuses on the available treatments for COVID-19, with a particular emphasis on antiviral drugs and host-directed therapies (HDTs). Antiviral drugs act directly against the virus and disrupt its ability to survive and/or replicate in the human body, while HDTs modulate the endogenous response (both excessive and deficient) to the virus, potentially leading to improved clinical outcomes. Owing to their broad mechanism of action, HDTs may also be effective against biological threat agents. Biological warfare agents may harm or incapacitate a very large number of human beings due to efficient infectivity and severity of disease [4]. In some cases, such as those of viruses listed as bioterror threats (https://emergency.cdc.gov/agent/agentlist-category.asp, accessed on 18 January 2023), there is no approved or efficient treatment to date [5]. In addition, a bioterror event may have significant national and international implications, similar to those observed during the COVID-19 pandemic.

This review aims to leverage the vast knowledge that has been amassed regarding COVID-19 to identify effective treatment options that are relevant as countermeasures to biological threat agents owing to their mechanism of action. We discuss the drugs that have been clinically approved for COVID-19 treatment, as well as those that have demonstrated potential effectiveness in advanced clinical trials. By doing so, we hope to contribute to the ongoing efforts to combat this disease and other potential biological threats, in particular biological threat agents sharing SARS-CoV-2 pathophysiological processes (e.g., penetration, replication, translation, transmission, infectivity and host responses).

## 2. Host-Directed Therapies (HDTs) for the Treatment of COVID-19

In the later stages of infectious diseases such as COVID-19, antimicrobial treatments may not be effective due to the fact that primarily damage at this stage is caused by the pathological disease responses rather than the pathogens. This is evidenced by the fact that there are often low titers of pathogen in the host during late-stage, critical disease. This is where host-directed therapies (HDTs) come into play, as they activate under-activated or suppress over-activated endogenous pathways to cope with a wide range of pathogens. HDTs have a lower risk of resistance developing by the pathogen compared to direct antiviral drugs, although they may have an increased risk of toxicity [6]. Throughout the COVID-19 pandemic, a variety of HDT were tested and are categorized in this review based on their mechanism of action, including (i) reducing overexcitation of the inflammatory response (Table 1); (ii) activation of the immune response (when depressed by the pathogen); and (iii) disruption of cellular mechanisms involved in viral life cycle and survival (Table 2). In the following passages, we elaborate on these three mechanisms.

### 2.1. Suppressing Over-Activated Immune Responses 

In patients with severe COVID-19, pro-inflammatory cells such as macrophages and neutrophils secrete high levels of inflammatory cytokines, leading to excess inflammation and potentially resulting in multi-organ failure and systemic collapse [7]. 

Hereinafter, we elaborate on HDTs aimed at reducing the overstimulation of the inflammatory response (“cytokine storm”) without compromising the proper immune system activity in combating the virus.

Several drug treatments have shown clinical benefits in COVID-19 patients by targeting the excessive immune response, including steroids, anti-cytokines, kinase inhibitors and selective serotonin reuptake inhibitors (SSRIs), as well as drugs that have undergone advanced clinical examination such as Sabizabulin, Vilobelimab, and Metformin.

#### 2.1.1. Steroids

##### Dexamethasone

Steroids have shown promise as potent anti-inflammatory agents in treating the “cytokine storm”, which, as has been mentioned above, is a hallmark of severe COVID-19. Dexamethasone, a steroid recommended for use in COVID-19 patients, has been shown to reduce mortality rates by approximately 33% in mechanically ventilated patients and about 20% in patients requiring noninvasive oxygen treatment. Recent studies have estimated that treatment with Dexamethasone saved the lives of about one million COVID-19 patients worldwide as of March 2021 [8]. 

##### Budesonide

A phase 2 clinical trial of early treatment with the inhaled steroid Budesonide in mild–moderate COVID-19 patients demonstrated effectiveness in reducing rates of emergency room visits and hospitalizations, shortening recovery time, and improving clinical symptoms [9]. 

These findings underscore the potential of steroid treatment at all stages of COVID-19 and other respiratory diseases caused by pathogens.

#### 2.1.2. Anti-Cytokines

Progressive elevation of various inflammatory cytokines, including Interleukin-6 (IL-6), IL-1β and granulocyte–macrophage colony-stimulating factor (GM-CSF), is directly associated with disease severity. Significantly higher serum levels of pro-inflammatory cytokines are observed in critically ill patients, prompting specific anti-cytokine treatment for prevention of tissue damage [10,11,12,13]. There are several possible treatments that suppress excessive inflammatory response, including the use of antibodies that specifically target different cytokines (or their receptors) to neutralize them and prevent their pro-inflammatory activities.

##### Monoclonal Antibodies Targeting IL-6 Receptor

Monoclonal antibodies against cytokine IL-6 have shown promise in mitigating the pathology and “cytokine storm” associated with severe COVID-19 [14]. 

Tocilizumab is a monoclonal antibody that targets the IL-6 receptor. This drug has demonstrated efficacy when administered to severely ill patients within two days of admission to an intensive care unit (a multicenter cohort study of 4485 patients conducted from March to May 2020). A significant reduction in mortality within 30 days was observed (27.5% in the treated group compared to 37.1% in the control group) [15]. A subsequent study involving over 4000 patients demonstrated a reduction in hospital stay time (as measured by the number of patients discharged within 28 days, 57% in the treated group compared to 50% in the control group), as well as a reduction in the likelihood of invasive mechanical ventilation and mortality (35% mortality in the treated group compared to 42% in the control group) [16]. In June 2021, the drug was granted emergency use authorization by the FDA for use in combination with steroids in hospitalized COVID-19 patients who require supplemental oxygen or mechanical ventilation (noninvasive, invasive and extracorporeal membrane oxygenation (ECMO)) (https://www.fda.gov/media/150319/download, accessed on 9 May 2023). This emergency use authorization was based, in addition to the aforementioned studies, on smaller clinical studies conducted on hundreds of patients (https://www.fda.gov/news-events/press-announcements/coronavirus-covid-19-update-fda-authorizes-drug-treatment-covid-19, accessed on 9 May 2023). On 21 December 2022, the drug (in combination with systemic steroids) was granted formal FDA approval (https://www.gene.com/media/press-releases/14979/2022-12-21/fda-approves-genentechs-actemra-for-the-, accessed on 9 May 2023).Sarilumab, another monoclonal antibody targeting the IL-6 receptor, has been found to be effective in reducing mortality rates in COVID-19 cases. In meta-analyses comparing Sarilumab to Tocilizumab, Sarilumab showed potential effectiveness, although Tocilizumab appeared to be more effective overall [17,18]. However, it is worth noting that a phase 3 clinical trial of Sarilumab did not demonstrate significant improvement compared to those who were not treated with the drug. It is important to keep in mind that only 60% of the study participants were treated with steroids [19].

##### Anakinra

Anakinra, a recombinant Interleukin-1 (IL-1) receptor antagonist, is a drug that inhibits the activity of cytokine IL-1. Several studies have shown that IL-1 plays a key role in the development of the “cytokine storm” in patients infected with SARS-CoV-2, and that the levels of the pro-inflammatory cytokine IL-1β increase significantly following infection [20]. In a phase 3 clinical trial involving 600 severe COVID-19 patients, treatment with Anakinra (administered in combination with the steroid Dexamethasone) resulted in a significant clinical benefit, including a significant decrease in mortality rates compared to the control group who received Dexamethasone alone. This effect was observed until day 28 from the start of treatment, along with a decrease in the length of hospitalization or stay in the intensive care unit. A systematic review and individual patient-level meta-analysis [21] of hundreds of patients revealed that Anakinra significantly lowers the risk of mortality on the 28th day in hospitalized patients in a moderate–severe condition who are not treated with Dexamethasone, especially when the CRP (C reactive protein) levels in the blood are above 100 mg/mL.

Observational and retrospective studies, as well as smaller clinical trials, have also demonstrated the efficacy of Anakinra in improving survival rates, preventing deterioration to pulmonary failure, and improving inflammation indicators in COVID-19 patients [22,23,24,25,26].

On 2 November 2020, Anakinra received emergency use authorization (EUA) from the FDA for the treatment of severe COVID-19 patients who are hospitalized with pneumonia, are in need of oxygen support, and are also characterized by elevated levels of soluble urokinase plasminogen activator receptor (suPAR) in plasma. The clinical efficacy of the drug treatment was demonstrated in a subgroup analysis and was maintained up to 90 days after treatment onset [27].

##### Infliximab

Infliximab is a monoclonal antibody against TNFα. When administered alone in a phase 3 clinical trial, a 41% relative reduction in measured mortality rates up to day 28 of treatment was observed (from 14.5% in the control group to 10.1% in the treatment group, 517 patients in moderate or severe condition). Chances for clinical status improvement were also higher (43.8% increase) up to day 14 from treatment start. The drug was administered in combination with steroids or Remdesivir [28].

##### Monoclonal Antibodies against Granulocyte–Macrophage Colony-Stimulating Factor

Monoclonal antibodies against granulocyte–macrophage colony-stimulating factor (GM-CSF) are being developed as anti-inflammatory therapies. GM-CSF is a cytokine that activates the development of immune system cells (granulocytes and macrophages) from precursor cells, and it has been implicated in the inflammatory processes. Antibodies against GM-CSF and its receptor have been developed for the treatment of chronic inflammatory diseases such as arthritis and asthma [29]. High levels of GM-CSF were associated with severe COVID-19 patients, which may distinguish these patients from those with seasonal influenza (in the latter case, elevated GM-CSF was not detected, in contrast to IL-6, which was elevated in both morbidities) [13].

Lenzilumab is a monoclonal antibody that targets GM-CSF. Results from a phase 3 clinical trial in approximately 500 critically ill patients (without “cytokine storm” or invasive mechanical ventilation, also treated with Remdesivir and/or steroids) show that treatment with the drug led to a statistically significant (*p* < 0.04), though mild (84% versus 78%) increase in survival rates in non-invasively mechanically ventilated patients up to day 28 of treatment compared to patients who did not receive Lenzilumab. However, according to a statement by Humanigen, the company which develops this drug, a separate clinical trial examining the effect of the drug in combination with Remdesivir (compared to Remdesivir alone) showed no benefit compared to treatment with Remdesivir alone (https://s28.q4cdn.com/539885110/files/doc_news/Humanigen-Receives-Preliminary-Topline-Data-From-NIHNIAID-Study-of-Lenzilumab-in-ACTIV-5BET-B-2022.pdf, accessed on 18 May 2023).Otilimab is another monoclonal antibody that targets and inhibits the activity of the cytokine GM-CSF. In a phase 2 clinical trial (OSCAR trial) in COVID-19 patients with systemic inflammation, Otilimab did not demonstrate efficacy in patients aged 18 and older, but there was a partial positive trend observed in the subgroup of patients aged 70 and older in terms of survival rates and decrease in inflammatory markers (this trend was not confirmed in a separate successive study; however, in this separate study, a reduction in inflammatory markers was observed with Otilimab, in addition to the establishment of an acceptable safety profile) [30]. While these results suggest some potential benefit of treatments that target the anti-GM-CSF mechanism in COVID-19 patients, further research is needed to confirm these findings.

#### 2.1.3. Kinase Inhibitors

##### Janus Kinase (JAK) Inhibitors

The JAK inhibitors Baricitinib and Tofacitinib are drugs used to treat joint inflammatory diseases by inhibiting the JAK1/2 and JAK1/JAK3 enzymes, respectively [31]. Imatinib inhibits the Bcr-Abl kinase by binding to its active site for Chronic Myeloid Leukemia (CML) treatment.

A meta-analysis of four controlled clinical trials (10,815 patients) showed that treatment with Baricitinib in hospitalized COVID-19 patients led to a significant decrease in a 28-day mortality, as well as a positive trend (although not statistically significant) in the reduction in invasive mechanical ventilation (IMV) or ECMO support [32,33]. The drug was approved for emergency use in hospitalized COVID-19 patients supported by noninvasive oxygen (or on mechanical ventilation) in October 2022 as a standalone treatment, following the previous approval of combination therapy with Remdesivir. The combination therapy of Baricitinib and Remdesivir led to a faster recovery compared to that of Remdesivir alone [34] and was characterized by a better safety profile compared to that of the Dexamethasone–Remdesivir combination [35].Similarly, treatment with Tofacitinib led to a significant decrease in mortality or in the development of respiratory failure (18.1% vs. 29% in the placebo group) within 28 days. Death from any cause through day 28 occurred in 2.8% vs. 5.5% of those in the Tofacitinib or placebo group, respectively (STOP-COVID trial [36]).

##### Imatinib

Imatinib has revolutionized the treatment of Chronic Myeloid Leukemia (CML) by dramatically improving patient outcomes. It works by binding to the active site of the Bcr-Abl kinase and blocking its activity, which in turn inhibits the proliferation and survival of leukemic cells. In the past, it was demonstrated that the drug leads to hemodynamic improvements in patients diagnosed with pulmonary arterial hypertension through protective effects on healthy blood vessels [37] by, among other things, inhibiting the influence of inflammatory cytokines such as IL-6 and TNFα [38]. Experiments in animal models indicate that Imatinib protects the endothelial barrier in inflammatory conditions and prevents the development of inflammation by inhibiting the Abl kinase [39,40]. Isolated clinical case reports have pointed to the potential for rapid correction of vascular leakage and improvement in clinical status [41], as well as a beneficial effect in COVID-19 patients [42]. A controlled clinical trial was conducted on COVID-19 hospitalized patients who required noninvasive oxygen support, which showed that the drug did not benefit in terms of achieving endpoint of the trial (48 h cessation of respiratory support). However, the mortality rates were lower in the Imatinib-treated group (8%) compared to the control group (14%), and the median time supported by mechanical ventilation was significantly lower in the treatment group (7 vs. 12 days) [43]. The improvement in survival was unadjusted, but a significant statistical effect was still observed even after adjustment. A follow-up study that examined the 90-day mortality rates from the start of treatment also showed a significant improvement in survival rates, with mortality rates of 9.1% and 16.5% in the treatment and control groups, respectively (which remained significant even after adjustment). In addition, the median time without a ventilator support (ventilator-free days) was 84 days in the treatment group compared to 64 days in the control group, and there was a significant decrease in the duration of invasive ventilation. Furthermore, the length of stay in the intensive care unit was shortened from 15 to 9 days following Imatinib treatment, and a relative improvement in clinical status was demonstrated over the 90-day follow-up period [44]. It was also suggested that the effect of Imatinib on mortality occurred due to the reversal of endothelial dysfunction rather than anti-inflammatory effects [45].

#### 2.1.4. Selective Serotonin Reuptake Inhibitors (SSRIs)

Fluoxetine/Fluvoxamine, used to treat depression, are medications that belong to the SSRIs family. These medications have an anti-inflammatory effect that is mediated through agonism to sigma-1 receptors (S1Rs), which leads to a decrease in cytokine production [46]. In a large, multi-center retrospective study that included 83,584 COVID-19 patients hospitalized in 87 medical centers in the United States, an inverse correlation was observed between the administration of these medications and severity of illness and death. This study was based on previous findings, including observational and preclinical findings (and even in vitro studies), suggesting that treatment with these drugs may be beneficial for treatment of COVID-19 [47,48]. Specifically, clinical studies, some of which are ongoing (although relatively small), have shown that treatment with Fluvoxamine reduces the risk of clinical deterioration in non-hospitalized patients [49], including early-stage treatment in the at-risk population [50]. It should be emphasized, though, that two large clinical trials failed to show a beneficial effect of Fluvoxamine treatment. However, one of these studies included only a vaccinated population [51], and in the other study, the median age was below 50 years, and only half of the participants were unvaccinated. In addition, the dosage chosen for treatment was lower than the dosages chosen for the trials described above [52]. In addition, treatment with oral Fluvoxamine plus inhaled budesonide among highly vaccinated, high-risk outpatients with early COVID-19 reduced the risk of deteriorating to severe disease in comparison to placebo-treated patients [53]. With respect to infections with the Omicron variant, according to a recently published study [54], antidepressant use as a whole (non-SSRIs, SSRIs and fluoxetine specifically) was associated with a lower risk of severe COVID-19 (ICU admission and inpatient death). This provides supportive evidence for the treatment potential of all antidepressants for severe COVID-19. Another study that has not yet undergone peer review [55] suggests that SSRIs with agonist activity at the sigma-1 may significantly reduce long-term complications of COVID-19 (post-acute sequelae of COVID-19, “Long-COVID”). Several mechanisms have been proposed so far for the beneficial effects of these drugs, including inhibition of the enzyme acid sphingomyelinase (ASM) and anti-inflammatory effects. ASM [56] is responsible for converting sphingomyelin into ceramide and phosphorylcholine. Ceramide is a central component of the cell membrane through which SARS-CoV-2 penetrates into the intracellular compartment. As for the anti-inflammatory activity, the drugs act on several levels: (i) through binding to S1Rs receptors, as mentioned, which leads to reduced activity of the endonuclease inositol-requiring enzyme 1 (IRE1) and decreased expression of cytokines without inhibiting classic inflammatory mechanisms; (ii) inhibition of classic inflammatory elements, such as nuclear factor κB, inflammasome, TLR4, and PPARγ; and (iii) inhibition of ASM on endothelial and immune system cells [57]. Additional mechanisms have been proposed to explain the beneficial effect of these drugs, such as anticoagulant/antiplatelet elements, direct antiviral elements and more [58].

#### 2.1.5. Sabizabulin

Sabizabulin is a chemotherapeutic drug currently undergoing clinical trials for the treatment of prostate cancer. It works through microtubule disruption, which is a key component of the cytoskeleton. Sabizabulin is bound to the active site of Colchicine (another drug that has shown potential as a treatment for COVID-19, but eventually failed to prove clinical efficacy [59]).

Compared to Colchicine, Sabizabulin binds with high affinity to the “Colchicine binding” pocket in the β-tubulin subunit, and has a superior pharmacokinetic profile, resulting in longer circulatory presence [60,61]. The development of Sabizabulin included in vitro experiments that demonstrated potential anti-inflammatory effects, although these data have not yet been published in the scientific literature (https://www.biospace.com/article/veru-s-lead-compound-shows-promise-as-antiviral-anti-inflammatory-therapeutic-for-covid-19, accessed on 18 May 2023). In a phase 3 clinical trial involving 204 participants (134 of whom were treated with Sabizabulin), a significant decrease in mortality rates (24.9% absolute reduction), a 43% relative reduction in ICU days (*p* = 0.0013), a 49% relative reduction in days on mechanical ventilation (*p* = 0.0013), and a 26% relative reduction in days in the hospital (*p* = 0.0277) were demonstrated in moderately to severely ill patients (with a high risk of ARDS and death) compared to the placebo group [62]. Despite these results, the FDA denied emergency use authorization (EUA) for the treatment of COVID-19 with Sabizabulin in early March 2023. 

#### 2.1.6. Vilobelimab

On the one hand, the complement system plays an important role in pathogen clearance. On the other, unregulated complement activation plays a crucial role in the pathogenesis of acute lung injury (ALI) induced by highly pathogenic virus including influenza A viruses H5N1, H7N9 and severe acute respiratory syndrome coronavirus (SARS-CoV-1). In particular, the C5a component of the complement system and its receptor, C5aR1, play an important role in inflammatory processes, particularly in tissue and blood vessel damage by recruiting neutrophils and monocytes to the site of inflammation. Blockade of C5a signaling has been implicated in the treatment of ALI induced by highly pathogenic viruses [63]. 

Vilobelimab is a monoclonal antibody (mAb) that targets C5a. During the COVID-19 pandemic, it has been shown that levels of C5a in COVID-19 patients are positively correlated with disease severity, and that C5aR1 antagonists prevent the development of acute lung injury in mice exposed to SARS-CoV-2 [64]. Results from a phase 3 clinical trial conducted in approximately 370 mechanically ventilated COVID-19 patients demonstrated that treatment with Vilobelimab in addition to standard care leads to a significant improvement in survival rates (a 31% mortality compared to 40% in the control group) [65]. On April 2023, it was declared that circumstances exist justifying the authorization of emergency use of Vilobelimab (https://www.fda.gov/media/166823/download, accessed on 18 May 2023).

#### 2.1.7. Metformin

Metformin is a drug used to treat type II diabetes mellitus (T2DM) with anti-inflammatory activity, including suppression of cytokine release (IL-6 and IL-1β), inhibition of inflammasome activity (including in mice, [66,67]), activation of autophagy, and more. The drug improved survival rates in mice in an acute lung injury model [68,69,70]. Observational retrospective studies and meta-analyses have shown a trend of decreased mortality, severity of illness, and hospitalizations following the use of this drug, mainly in patients with T2DM [70,71,72,73,74,75]. A retrospective analysis of approximately 6000 COVID-19 patients demonstrated a significant decrease in mortality in women (but not in men) with T2DM or obesity who were treated with Metformin [76]. Since T2DM is a major risk factor affecting prognosis in COVID-19 patients, alongside other risk factors that are associated with this vulnerable population (such as age and obesity) [77,78], it is important to examine in depth whether the anti-inflammatory effects or the anti-diabetic properties of the drug are the leading factors contributing to its beneficial effects. A controlled clinical trial (including approximately 1450 COVID-19 patients without diabetic background) showed no clear effect of Metformin on endpoints. However, a potential effect in preventing more severe components (such as intensive care unit admission, hospitalization, and mortality) was demonstrated. It should be noted that the study population was selected based on weight (all had obesity) and had a relatively young median age (46). Moreover, the hospitalization and mortality rates were minimal in this study. In addition, half of the study participants were vaccinated [52]. Therefore, it is possible that these data partly reflect a more pronounced effect of the drug. In another controlled study that examined early treatment with Metformin in non-hospitalized patients, the drug’s efficacy was not demonstrated in terms of hospitalization rates within 28 days of treatment initiation, as well as viral clearance and mortality [79]. However, Metformin administration to the same sub-population (early outpatient treatment) resulted in a 41% relative decrease and a 4.1% absolute decrease in the Long-COVID (≥9 months after infection) incidence (results taken from a phase 3 trial [80]). This metformin effect was consistent across subgroups, including viral variants.

#### 2.1.8. Abatacept

Abatacept is an analogue of CTLA-4 that prevents the generation of a co-stimulatory signal in antigen-presenting cells, thereby preventing overactivation of the immune system. Treatment with the drug as a single dose led to a decrease in mortality rates documented up to day 28 after treatment: 11% in the treatment group (509 patients with moderate or severe conditions) as opposed to 15.1% in the control group (a decrease of 38%, odds ratio of 0.62) [81]. At the pandemic onset, it was hypothesized that treatment with Abatacept, which blocks the pro-inflammatory axis of the CD80/86 co-stimulatory signal, could be an effective approach in COVID-19 patients; this due to the suppression of the “cytokine storm”, particularly IL-6 levels (which are highly correlated with severe disease). Additionally, the drug is expected to suppress the complement system and the overactivation of the immune response mediated by B lymphocytes [82].

### 2.2. Immune Response Stimulation

The immune response of the host is a critical factor in determining the pathogenicity of viruses and their ability to cause disease. COVID-19 patients experience severe illness due to the suppression of immune activity at the onset of the disease and excessive activity in its later stages. The SARS-CoV-2 virus is known to evade the host’s defense mechanisms through various mechanisms such as promoting programmed cell death of natural killer cells (NK cells) and delaying the production of interferon-α and-β [7,83]. In this section, we explore several therapeutic strategies to induce immune responses effective in combating the virus (Table 2, upper panel). These strategies are based on studies that have shown the potential of stimulating the activity of the immune system which can serve as an important treatment for COVID-19 patients.

#### 2.2.1. Interferons

The interferon response is the first line of defense against viruses. Detection of viruses by the innate immune system leads to the production of type I (IFNα, IFNβ) and III (IFNλ) interferon responses [84]. Accordingly, one mechanism viruses use to evade or suppress the immune response is by inhibiting the interferon response, as observed for SARS-CoV-2 [85]. An external activation of the immune response by the administration of exogenous interferons is a well-known approach for combating viruses such as Hepatitis B, Hepatitis C, and HIV-1 [86]. It should be noted that interferon therapy is not without side effects due to the inflammatory response and cytokine release downstream of interferon signaling, as observed for type I interferons [87], which requires careful dosage adjustment. In contrast, type III interferon response does not promote inflammation [88], and therefore may have a more favorable safety profile. On top of this, a study that examined the immune response in COVID-19 patients treated with IFNλ found that the treatment lead to a boosted innate response without negatively affecting the adaptive immune response [89]. Additionally, in a mouse model, full protection was achieved against lethal exposure to hCoV and MERS-CoV following treatment with IFNλ (prophylactic or post-infection) [90].

The suppression of the interferon response in COVID-19 patients, alongside the potential therapeutic value of IFNs, has led to pre-clinical and clinical studies to evaluate the effectiveness of this treatment. Some of the clinical trial results are indeed very encouraging, particularly in terms of therapeutic intervention in the early stages of the disease, and in particular regarding IFNλ.

##### Pegylated IFNλ-1a

The administration of Pegylated IFNλ-1a has previously been examined for the treatment of Hepatitis B/C/D. A single subcutaneous injection within 7 days of symptom onset led to a 50% reduction (2.7% compared to 5.6% in the control group) in hospitalizations and emergency room (ER) admissions by day 28 of treatment, as well as a 60% decrease in mortality. This was demonstrated in a phase 3 clinical trial in non-hospitalized patients (approximately 900 patients received Pegylated IFNλ-1a out of the 1900 patients enrolled), the majority of whom (84%) were vaccinated against COVID-19. The treatment was effective for infection with various SARS-CoV-2 variants, including Omicron [91].

##### IFN β-1a

SNG001, IFN β-1a, which is administered by nebulization to hospitalized patients for 14 days, demonstrated therapeutic potential (rapid recovery and higher likelihood of clinical improvement) alongside a positive safety profile in a phase 2 clinical trial of about 100 participants (half of whom received SNG001) [92]. Although the final and decisive stage of the trial did not lead to similar conclusions, it cannot be ruled out that this treatment might be more effective if administered at different time points (e.g., early treatment or prophylactic treatment) or in combination with other drugs. Additionally, it should be examined whether SNG001 has potential in treating other viral agents. 

#### 2.2.2. Nitazoxanide

Nitazoxanide, an anti-parasitic drug from the thiazolide family, activates the innate immune system by upregulating the expression of type I IFN pathway genes TLR7 and TLR8. These are involved in antiviral activity against various viruses [93,94]. Additionally, the drug has anti-bacterial activity [95,96]. A randomized clinical trial in approximately 400 mild COVID-19 patients who received Nitazoxanide for up to three days from symptom onset for a total of five days showed that the drug led to faster virus clearance (29.9% negative swabs in the study group versus 18.2% in the control group on day 5 of treatment) and a significant reduction in viral load [97]. In another experiment (with only 50 hospitalized mild COVID-19 patients), the treatment led to a significant decrease in hospitalization time (6.6 days in the study group versus 14 days in the control group), a decrease in inflammatory markers (including TNFα and IL-6), and an increase in negative PCR tests on day 21 after treatment initiation [98]. However, treatment with Nitazoxanide in hospitalized COVID-19 patients who required oxygen support did not lead to a decrease in mortality or clinical deterioration rates (defined as transfer to an intensive care unit). Nevertheless, a benefit was observed in terms of a shorter hospitalization time, clinical improvement, decreased need for oxygen, and decreased inflammatory markers on day 7 after treatment initiation [99]. In addition to activating the innate immune system and the anti-inflammatory effects of the drug (also demonstrated in preclinical models [100]), Nitazoxanide mediates cellular antiviral effects via phosphorylation of eukaryotic translation initiation factor 2-α (eIF2α) [101]. In light of the improvement that was achieved following early rather than late treatment, we believe that the favorable effects of the drug in COVID-19 was achieved mainly due to the stimulation of the immune response.

### 2.3. Disruption to Cellular Mechanisms Involved in the Viral Life Cycle and Survival

Upon entering a host, the virus initiates its replication process by binding to cells with appropriate receptors, such as pneumocytes and enterocytes. SARS-CoV-2 enters the infected cell through endocytosis, mediated by the spike protein of the virus that binds to the angiotensin-converting enzyme 2 (ACE2) receptor on the hosts’ cells. Following endocytosis, the virus releases its genome into the host cell cytoplasm, initiating the translation process for viral protein synthesis. These proteins assist in the replication of the viral genome and the creation of viral envelopes. Ultimately, all viral components assemble into complete virions that are released from the host cell through the process of exocytosis. The life cycle of the virus can be divided into three main stages: entry, genome replication, and release from the host cell [7,102]. In the following section, we elaborate on drugs that disrupt cellular mechanisms involved in viral life cycle and survival (Table 2, lower panel).

#### 2.3.1. Plitidepsin

Plitidepsin is a chemotherapeutic drug used to treat plasma cell disorders (multiple myeloma). Plitidepsin inhibits the eukaryotic translation-elongation-factor-1A (eEF1A), an endogenous protein involved in the translation process of viral and cellular proteins. The drug was shown to inhibit viral replication, including in mice infected with SARS-CoV-2. It should be noted that the antiviral activity of the drug against SARS-CoV-2 is 27.5 times more potent than that of Remdesivir [103,104], and relevant clinical findings suggest potential effectiveness [105]. The drug is currently undergoing clinical evaluation (phase 3) in combination with Dexamethasone in approximately 610 patients (ClinicalTrials.gov, NCT04784559). In terms of the drug’s potential as a broad-spectrum antiviral treatment, it has been shown in the past that eEF1A plays an important role in the replication of RNA viruses [106,107], including influenza A [108] and respiratory syncytial virus (RSV) [109]. Additional research indicates that the interaction of eEF1A with a specific RNA segment (3′ (+) stem-loop RNA) of West Nile virus accelerates virus synthesis [110]. This interaction is relevant for all Flaviviridae virus family members.

#### 2.3.2. HDTs with Antiviral Activity

For some of the drugs mentioned above (demonstrated as having potential efficacy against COVID-19 by reducing excessive inflammatory response), an additional mechanism of direct antiviral activity is involved by interfering with cellular mechanisms involved in the viral life cycle and survival.

##### Baricitinib

In addition to its anti-inflammatory activity discussed earlier, Baricitinib acts directly by potentially inhibiting numb-associated kinases (AAK1, GAK, BIKE, and STK16) that influence AP-2, a scaffolding protein vital for the entry and proliferation of the virus into cells [111]. It has been previously shown that inhibiting this kinase family leads to decreased infectivity of various viruses, including Dengue, SARS-CoV-1, and Ebola [112].

##### Sabizabulin

Direct antiviral activity is attributed to microtubule (MT) inhibition as MTs play an important role in intracellular transport of virus particles and regulation of other processes [113]. The importance of microtubules in viral attachment and intracellular transport has been demonstrated for RNA and DNA viruses. Microtubule inhibitors have also shown direct antiviral activity against SARS-CoV-2 [114].

##### Imatinib

In addition to the immunomodulatory activity mentioned above, Imatinib has shown antiviral activity (inhibition of virus replication) against MERS and SARS-CoV-1 in the early stages of infection, after internalization and endosomal trafficking, by inhibiting viral fusion to the endosome membrane [115]. Specific studies on SARS-CoV-2 have shown direct activity on the ACE2 receptor (enzymatic activation and allosteric inhibition of ACE2 binding to spike protein), as well as inhibition of viral entry into cells (endocytosis- and membrane-fusion routes) [116,117].

##### Metformin

In addition to its suppressing the over-activated immune response (discussed earlier), a direct antiviral effect was demonstrated for Metformin via 5′-AMP-activated protein kinase (AMPK) activation [118]. An antiviral effect of the drug was demonstrated against various viruses, i.e., Zika, Dengue [119] and Hepatitis C [120].

## 3. Antiviral Treatments against SARS-CoV-2

### 3.1. Protease Inhibitors

As the pandemic progressed, several antiviral drugs have been developed that have demonstrated effectiveness in treating COVID-19 patients. One of the most effective drugs is Paxlovid (Nirmatrelvir/Ritonavir) [121], which is administered orally and inhibits the main protease (Mpro) of the virus [122]. However, since Paxlovid specifically targets human coronaviruses, it apparently has no therapeutic potential against other viral pathogens known as BWA and is therefore not discussed in this review (nor are other advanced development drugs that inhibit SARS-CoV-2 Mpro).

Unlike Paxlovid, several broad-spectrum antiviral drugs that have demonstrated clinical efficacy against SARS-CoV-2 may serve as a therapeutic arsenal against viruses on the bio-threat list. 

### 3.2. Remdesivir

Remdesivir is a broad-spectrum antiviral drug. This drug was the first to be approved as a treatment for COVID-19. Remdesivir is a prodrug of the adenosine analog that inhibits the viral RNA-dependent RNA polymerase (RdRp) enzyme [123]. The drug is administered intravenously to non-hospitalized patients with mild-to-moderate symptoms who are at risk (patients with comorbidities), as well as in combination with other drugs in hospitalized patients. The drug’s development began years before the coronavirus pandemic outbreak, and it demonstrated efficacy against the Ebola virus in preclinical models (primates) [124]. Initial results in humans also appeared promising [125]; however, the drug’s usefulness was demonstrated to be lower than that of monoclonal antibodies against the virus. This led to the cessation of the clinical trial evaluating its efficacy in humans [126].

### 3.3. Molnupiravir

Molnupiravir is an oral antiviral drug that has been clinically approved for the treatment of COVID-19. Like Remdesivir, it works by inhibiting the RdRp enzyme. The drug has lower efficacy compared to Paxlovid for treating COVID-19 (leading to an only about 30% reduction in mortality and hospitalization rates) [127], but unlike Paxlovid, it may be effective against a wide range of viruses, including influenza and Venezuelan equine encephalitis virus (for which it was originally developed in 2013) [128,129]. Additionally, the drug has a favorable safety profile and is characterized by minimal drug interactions (a crucial aspect for the population at large). Findings from preclinical models of SARS-CoV-2 in hamsters indicate a significant reduction in transmission rates [130], suggesting the drug’s potential to interrupt transmission chains during an outbreak.

### 3.4. Inhaled Nitric Oxide

Nitric Oxide (NO) is a gaseous signaling molecule involved in many physiological and pathological processes. In particular, NO has antimicrobial effects in vitro and in vivo [131,132]. It has also been shown to have antiviral effects on the SARS-CoV-1 virus, namely inhibition of virus replication cycle and direct inhibition of RNA synthesis in cells [133]. While respiratory treatment (inhaled NO, iNO) did not show effectiveness in patients in critical condition [134,135], a phase 3 clinical study in approximately 300 symptomatic patients in the early stage of the disease indicated that a one-week treatment with an NO-releasing nasal spray leads to a significant reduction in viral load and a 4-day shortening of time till virus clearance [136]. Another study in approximately 90 mild symptomatic patients showed similar results, with a 16-fold reduction in viral loads within 2–4 days and faster symptom resolution. The clearance may explain the faster disappearance of symptoms and may be correlated with lower chances of infection and lower probability of hospitalization or deterioration to severe disease [137]. It should be noted that NO has anti-inflammatory effects and may act as an anti-thrombotic agent and vasodilator [138], but given the route of administration and early stage of treatment, we estimate NO’s efficacy is exerted through a direct antiviral mechanism.

## 4. Effective COVID-19 Treatments and Their Potential Efficacy towards Biological Threat Agents

Treatments with immunomodulatory drugs that affect the activity of the immune system should be applied with great caution due to the critical role of the immune system in coping with pathogens and the increased risk of adversely impacting the course of infections [139,140]. However, it should be emphasized that combining immunomodulatory drugs with antiviral or antibacterial drugs may improve clinical outcomes and survival following exposure to biological threat agents. 

It is important to emphasize that the antimicrobial activity of the drugs mentioned in this section is addressed as a platform for discussion purposes only. In case of need, each drug should be examined for its potential effectiveness against different biological threat agents, including conducting clinical trials, dose adjustments and a formal and specific use approval by the proper authorities.

Combining antibodies directed against Y. pestis with methylprednisolone in mice exposed subcutaneously to Y. pestis improved survival and inflammatory markers tested [141]. In a pneumonic plague model, pre-exposure treatment with the inhaled steroid Fluticasone followed by late antibiotic treatment also led to decreased inflammatory markers and increased survival rates [142]. In addition, treatment with steroids [143] or Anakinra [144] in pre-clinical models of respiratory exposure to ricin led to increased survival rates and improved pathology (in the case of toxins, where the damage is sterile, there is no concern regarding immune modulation and infectivity). Combination therapy of Etanercept, a monoclonal antibody against TNFα, with the antiviral drug Cidofovir, led to clinical improvement and increased survival in mice following respiratory infection with the Ectromelia virus (mousepox, an animal model of smallpox) [145].Some of the drugs discussed in this review have direct or indirect antimicrobial activities, such as Metformin, which has been shown to have antiviral [120,146] and antibacterial [147] activity when used in combination with other drugs. Metformin’s effectiveness against flavivirus (Zika and Dengue viruses) replication has been demonstrated in cells [148]. SSRIs, such as Fluoxetine, also have direct antibacterial activity and synergistic activity with various antibiotics [149]. Furthermore, Fluoxetine significantly inhibited cell infection with the Ebola virus through inhibition of the enzyme acid sphingomyelinase [150].The drug Nitazoxanide has been shown to have antibacterial effects in combination with antibiotics [151] as well as broad-spectrum antiviral effects [93], including against coronaviruses [152] and various other pathogens on the bio-threat list [93].Concerning biological threat viral agents, kinase inhibitors, including Imatinib, Baricitinib, and Tofacitinib, are at different stages of clinical and preclinical research to evaluate their effectiveness as broad-spectrum antiviral treatments [153].

## 5. Summary

In this review, we expanded on therapeutic treatments that have been tested clinically in COVID-19 patients and demonstrated efficacy. We expanded on HDTs, as well as on broad-spectrum antiviral drugs.

It is important to emphasize that the effectiveness of the treatments mentioned above is context dependent. Efficacy varies, depending on the stage and severity of the disease. Some drugs were suggested only for late stages of the disease in severe patients (such as Dexamethasone), while others were only helpful when administered in early stages (such as Pegylated IFNλ-1a, Nitazoxanide and iNO). Anakinra, for example, has four parameters influencing therapeutic efficacy: duration of treatments (optimally ≥ 10 days), dose (≥100 mg), administration route (intravenous) and early initiation of therapy [154].

Some drugs showed efficacy only when administered in combination with another drug (such as Tocilizumab or Anakinra, effective with Dexamethasone). Baricitinib was initially approved as part of a combination treatment, and only later approved as a stand-alone treatment.

A class effect could not be implemented automatically, as there was often a difference in efficacy between drugs belonging to the same family. For example, in some JAK inhibitors, no efficacy or only partial efficacy was observed. This could be due to differences in their activity against different JAK isomers, differences in safety and pharmacokinetic data, or the presence of direct antiviral elements that exist in parallel to the anti-inflammatory activity only in certain drugs. Another example is SSRIs, whose effectiveness in COVID-19 patients was mainly demonstrated in sigma-1 receptor agonists.

HDTs have prominent advantages over drugs with direct antimicrobial activity, allowing their use for dealing with a variety of pathogens:A broad coverage range: viruses, bacteria, toxins and more due to the endogenous nature of the effect.Reduced risk of developing microbial resistance (namely the increased potential of coverage of variants, antibiotic resistant bacteria and more).

Obviously, this does not contradict the possibility of combining HDT with drugs with direct antimicrobial activity (as well as combining different HDTs operating via different mechanisms).

Due to the exceptional efficacy of both mRNA vaccines for COVID-19 and the approved antiviral drugs for treating COVID-19 disease, motivation for relevant clinical trials has significantly declined. Nonetheless, we believe it is still essential to monitor ongoing research in the following areas of emerging modalities of medicine and advanced therapeutic approaches:Mesenchymal stromal cells (MSCs), which have demonstrated potential efficacy in both preclinical models and clinical settings against infectious diseases (bacterial, viral, etc.) via direct antimicrobial mechanisms or host-directed therapy (HDT). However, MSCs also face several challenges such as safety and regulatory issues [155]. Clinical trials with MSCs in COVID-19 patients, as they become available [156], may help address these gaps in safety, efficacy, and regulation, potentially rendering this platform more applicable in the foreseeable future. MSCs mode of action may very well be relevant to many other microbial infections.Drug delivery systems and novel formulations are expected to streamline and optimize therapeutic treatments. Among other objectives, these systems aim to enable more convenient treatments (e.g., inhaled, per os or ambulatory-based administration versus the current parenteral administration in hospitals), reduce drug side effects (through local and organ-targeted delivery) and/or improve bioavailability. For example, a phase 1 clinical trial has been conducted to test an inhalable or intranasal formulation of the drug Niclosamide. Niclosamide is a drug possessing 40 times the potency of Remdesivir in Vero cells against SARS-CoV-2. However, it is not adequately absorbed when administered to COVID-19 patients [157].

All of the above is relevant for coping with biological threat agents. The drugs reviewed here may be repurposed to serve as a therapeutic arsenal for dealing with these agents, which are extremely deadly pathogens with the potential to cause mass casualties. Some of these pathogens are devoid of approved treatment, while for others standard treatment options leave room for improvement.

The current pandemic has led to significant investment in resources and budgets to develop diverse medical treatments, technologies and devices, resulting in an unprecedented pace and volume of clinical trials. This was fueled by both the medical urgency and government participation in funding and fast-forwarding the research. Like many challenging emergencies of the past, the pandemic has catalyzed substantial progress in developing, testing and approving therapeutics that, under normal circumstances, would have taken much longer to achieve. The enormous amount of data derived from so many clinical trials is a once-in-a-lifetime opportunity to establish broad-spectrum antimicrobial treatments that would be suitable for diverse infectious diseases, in particular diseases derived from exposure to biological threat agents. Every possible clinical data point should be utilized for preparing and dealing with future outbreaks.

Such events would require treatment protocol adjustments, tailored pharmaceutical dosing and relevant drug–drug interaction studies. Obviously, none of the above negates the need for developing in-depth understanding of the pathogen and disease, allowing the dissection of which clinical state is most relevant for treatment and which of the numerous avenues described above (as well as possible future additions) are suitable for the specific outbreak.

## Figures and Tables

**Table 1 microorganisms-11-01577-t001:** Host-directed therapies with beneficial effects against COVID-19.

Drug Class	Drug	Main Clinical Indication	Anti-COVID-19 Mechanism	Clinical Phase (COVID-19) [#Clinical Trial]	COVID-19 Indication
Steroids	Dexamethasone	Inflammatory and Autoimmune diseases	Anti-inflammatory	Recommended by the COVID-19 Treatment Guidelines Panel [NCT04381936]	hospitalized patients who require supplemental oxygen
	Budesonide	Airway diseases		2 [NCT04416399]	mild–moderate COVID-19 patients
Anti-cytokines	Tocilizumab	RA	Anti-IL-6 receptor	Approved [NCT04381936]	hospitalized patients above 2 years of age receiving systemic corticosteroids and requiring supplemental oxygen, NIMV or IMV, or ECMO
	Sarilumab	RA	Anti-IL-6 receptor	Recommended by the COVID-19 Treatment Guidelines Panel [NCT04315298; NCT02735707; NCT04327388]	recommended only when Tocilizumab is not available or feasible to use
	Anakinra	RA	Recombinant IL-1 receptor antagonist	EUA [NCT04318366; NCT04357366; NCT04680949; CRD42020221491]	hospitalized adults with pneumonia requiring supplemental oxygen (low- or high-flow oxygen) who are at risk of progressing to severe respiratory failure and likely to have an elevated plasma suPAR
	Infliximab	Autoimmune diseases (Arthritis, IBD)	Anti-TNFα	3 [NCT04593940]	moderate or severe condition in combination with steroids or Remdesivir
	Lenzilumab	Hematologic malignancies	Anti-GM-CSF	3 [NCT04351152; NCT04583969]	critically ill patients without “cytokine storm” or IMV, also treated with Remdesivir and/or steroids *
	Otilimab	Autoimmune diseases	Anti-GM-CSF	2 [NCT04376684]	a partial positive trend in patients above 70 years of age with systemic inflammation **
Kinase inhibitors	Baricitinib ***	RA	JAK1/2 inhibitor	EUA [NCT04401579; NCT04640168]	hospitalized patients requiring supplemental oxygen, IMV, or ECMO in combination with Remdesivir
	Tofacitinib ***	RA	JAK1/3 inhibitor	3 [NCT04469114]	recommended only when Baricitinib is not available or feasible to use
	Imatinib ***	CML	Abl kinase inhibitor	[EudraCT 2020-001236-10]	hospitalized patients who require noninvasive oxygen support
SSRIs	Fluoxetine/ Fluvoxamine ***	Depressive and Anxiety disorders	ASM inhibitors S1Rs agonists Anti-inflammatory Anticoagulant	2–3 [NCT04342663; NCT04727424; NCT05890586; NCT04510194]	early stage, non-hospitalized patients
Microtubule disruptors	Sabizabulin ***	Prostate cancer	Anti-inflammatory	3 [NCT04842747]	moderately to severely ill patients (with a high risk of ARDS and death)
Complement inhibitors	Vilobelimab	COVID-19	Anti-C5 mAb	EUA [NCT04333420]	hospitalized adults, initiated within 48 h of receiving IMV or ECMO
Biguanide anti-hyperglycemic	Metformin ***	T2DM	Anti-inflammatory autophagy	3 [NCT04510194; NCT04727424]	early outpatient treatment (women with T2DM or obesity)
CTLA-4 analogs	Abatacept	RA	Anti-inflammatory	3 [NCT04593940]	moderate or severe

ARDS—acute respiratory distress syndrome; ASM—acid sphingomyelinase; CML—Chronic Myeloid Leukemia; CTLA-4—cytotoxic T-lymphocyte-associated protein 4; ECMO—extracorporeal membrane oxygenation; EUA—Emergency Use Authorization; GM-CSF—granulocyte–macrophage colony stimulating factor; IBD—inflammatory bowel diseases; IL—interleukin; IMV—invasive mechanical ventilation; JAK—anus kinase; mAb—monoclonal antibody; NIMV—non-invasive mechanical ventilation; RA—rheumatoid arthritis; S1R—sigma-1receptors; SSRIs—selective serotonin reuptake inhibitors; T2DM—type II diabetes mellitus; suPAR—soluble urokinase plasminogen activator receptor; * demonstrated positive results; however, in a separate trial the drug did not demonstrate superiority over Remdesivir alone; ** result was not confirmed on a separate successive study; ^***^ these drugs also possess direct antiviral effects.

**Table 2 microorganisms-11-01577-t002:** Host-directed therapies stimulating immune response or disrupting viral life cycle with beneficial effects against COVID-19.

Antiviral Mechanism	Drug	Main Clinical Indication	Cellular Antiviral Target	Clinical Phase (COVID-19) [#Clinical Trial]	COVID-19 Indication
Immune response stimulation	Pegylated IFNλ-1a	Antiviral	N/A	Phase 3 [NCT04727424]	non-hospitalized patients
	IFN β-1a *	Antiviral	N/A	Phase 2 [NCT04385095]	hospitalized patients
	Nitazoxanide **	anti-parasitic	N/A	Phase 2 [NCT04552483; NCT04348409; NCT04561219]	mild hospitalized patients
Disruption of cellular mechanisms involved in viral life cycle and survival	Plitidepsin	Multiple Myeloma	eEF1A	Phase 3 [NCT04784559]	adult hospitalized patients with moderate COVID-19 infection (in combination with Dexamethasone)
	Baricitinib **	RA	NAKs	EUA [NCT04401579; NCT04640168]	hospitalized patients requiring supplemental oxygen, IMV, or ECMO in combination with Remdesivir
	Sabizabulin ***	Prostate cancer	MT inhibitor	Phase 3 [NCT04842747]	moderately to severely ill patients (with a high risk of ARDS and death)
	Imatinib ***	CML	ACE2	[EudraCT 2020–001236–10]	hospitalized patients who require noninvasive oxygen support
	Metformin ***	T2DM	AMPK	Phase 3 [NCT04510194; NCT04727424]	early outpatient treatment (women with T2DM or obesity)

ARDS—acute respiratory distress syndrome; ACE2—angiotensin-converting enzyme 2; AMPK—AMP-activated protein kinase; CML—Chronic Myeloid Leukemia; eEF1A—eukaryotic translation-elongation-factor-1A; ECMO—extracorporeal membrane oxygenation; EUA—Emergency Use Authorization; IMV—invasive mechanical ventilation; IFN—interferon; MT—microtubule; NA—not applicable; NAKs—numb-associated kinases; RA—rheumatoid arthritis; T2DM—type II diabetes mellitus; * efficacy was not demonstrated at the final stage of the trial; however, this treatment could be efficient if administered at early stage of the disease, and/or in combination with other drugs; ** Nitazoxanide also possesses anti-inflammatory and direct anti-viral effects; treatment of patients who required oxygen support did not lead to a decrease in mortality or clinical deterioration rates; *** this drug was mentioned above as host-directed therapy.

## Data Availability

The data presented in this review is contained within the article or cited literature.

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
