# Peer review of "Clinically Evaluated COVID-19 Drugs with Therapeutic Potential for Biological Warfare Agents"

_microorganisms, 2023, doi:10.3390/microorganisms11061577_

Round 1

Reviewer 1 Report

The authors summarized the approved drugs that might benefit COVID-19 patients. The literature review of this study are clear to convince the conclusion. The overall writing is also very clear. So, I think this study should be published in this specific issue. Still, I do have a few recommendations to strengthen the article: 

1) In Table 1 and 2, the authors should incorporate the clinical trial numbers in additional column.

2) SARS-CoV-2 attacks respiratory system, therefore the antivirals should accumulate in lung. The antivirals distribution should be discussed.

Author Response

Dear Reviewer,

We would like to thank you for comprehensive review of the manuscript and your suggestions for corrections.

According to your recommendations there is no doubt you are an expert in pharmacology. 

  1. Regarding the first suggestion, we added the clinical trial numbers to the Tables (corrections in the Tables were made in 'Track Changes'), it was an excellent idea.
  2. Regarding the second suggestion:
  • We previously considered discussing PK parameters of all drugs included in this review (before preparing the manuscript) and concluded that it could shift the focus of this review, that is already abundant in details (one exception was referring to the difference between colchicine and sabizabulin, since PK parameters may explain why colchicine did not work, in contrast to sabizabulin).
  • Another aspect of not mentioning lung accumulation was that, we think these drugs could be relevant to systemic exposure, i.e. to Bio-Threat agents (namely, emphasizing the lungs as a target organ doesn't serve the bottom line of this manuscript). 
  • In respect to your remark regarding accumulation of the anti-SARS-CoV-2 antivirals (remdesivir and molnupiravir) in lungs, to our opinion, it goes without saying that their pulmonary distribution is sufficient, since clinical efficiency was demonstrated in this well-defined pulmonary infection (COVID-19).
  • In light of this, we prefer not to discuss distribution of drugs to the lungs in our manuscript.

We hope you share with us the belief that, now, after correcting the Tables, this manuscript is well suited for publication in Microorganisms.

Sincerely,

Yoav Gal, Ph.D

Department of Biochemistry and Molecular Genetics

The Israel Institute for Biological Research

Ness-Ziona, Israel Email: [email protected]

Reviewer 2 Report

I have had the privilege of reviewing the manuscript titled "Clinically evaluated COVID-19 drugs with therapeutic potential for biological warfare agents" I must say that this manuscript provides a highly informative and insightful analysis of the current state of pharmacological-based treatments for COVID-19. The authors have done an excellent job of summarizing the key challenges and opportunities in combating the severe acute respiratory syndrome coronavirus 2 (SARS-CoV-2) outbreak.

The manuscript's structure is clear and well-organized, making it easy to follow the authors' arguments and analysis. The language used is scientific and concise, ensuring clarity while maintaining the depth of the subject matter. Additionally, the incorporation of recent literature on drugs under advanced clinical evaluation for COVID-19 with broad-spectrum activity demonstrates the authors' commitment to providing a comprehensive review.

In conclusion manuscript offers a well-rounded analysis of the current landscape of pharmacological interventions for COVID-19. Its exploration of host-directed therapies and their potential implications for public health preparedness is particularly commendable. I wholeheartedly recommend this manuscript for publication, as it contributes significantly to our understanding of the potential therapeutic strategies for COVID-19 and related respiratory infections, ultimately paving the way for improved patient outcomes and global health preparedness.

Author Response

Dear Reviewer,

We would like to thank you for thoroughly and comprehensively reviewing our manuscript entitled: “Clinically evaluated COVID-19 drugs with therapeutic potential for biological warfare agents”.

There is no doubt you are an expert in the field, and we have much appreciation for your warm wards and recommendation to accept our manuscript for publication in Microorganisms. It is our privilege to receive such an acknowledgement, which is not obvious to us.

Sincerely,

Yoav Gal, Ph.D

Department of Biochemistry and Molecular Genetics

The Israel Institute for Biological Research

Ness-Ziona, Israel Email: [email protected]
